

# Muscle contraction type-specific association of acceleration and deceleration performance with rates of force development

Hikaru Kurosaki[1,2], Ema Tsubota[1], Motoki Katsuge[1], Kosuke Hirata[3] and Kuniaki Hirayama[2]

[1] Graduate School of Sport Sciences, Waseda University, Tokorozawa, Saitama, Japan
[2] Faculty of Sport Sciences, Waseda University, Tokorozawa, Saitama, Japan
[3] Institute of Health and Sport Sciences, University of Tsukuba, Tsukuba, Ibaraki, Japan

Corresponding author
Kuniaki Hirayama,
k.hirayama@waseda.jp

## ABSTRACT

**Background:** Many sports require rapid acceleration and deceleration, particularly when changing direction. These movements require a large impulse, highlighting the importance of rates of force development (RFDs). However, the relationships between acceleration and deceleration performance and concentric and eccentric RFDs have remained uncertain. This study evaluated the correlation between RFDs in different muscle contraction types and acceleration and deceleration performances.

**Methods:** This study included 28 healthy subjects (13 males and 15 females; age: $21 \pm 2$ years; height: $1.66 \pm 0.09$ m; body mass: $65 \pm 10$ kg). Concentric, eccentric, and isometric RFDs were evaluated by having the subjects perform squat jumps, countermovement jumps, and isometric squats, respectively. Acceleration and deceleration performances were measured using a 10-yard (9.14 m) sprint and change of direction deficit ($COD_{deficit}$; calculated by subtracting the linear sprint time from the total time of the pro-agility test), respectively. Correlation analyses were performed to determine the relationship between the RFDs and the 10-yard sprint time and $COD_{deficit}$. The Pearson product-moment correlation coefficient ($r$) was used for normally distributed dependent variable combinations, whereas the Spearman rank correlation coefficient ($r(s)$) was applied when at least one variable was not normally distributed.

**Results:** A faster 10-yard sprint time was only correlated with greater concentric RFD ($r(s) = -0.41$, $p = 0.03$, 95% CI [$-0.69$ to $-0.03$]), whereas a smaller $COD_{deficit}$ was only correlated with greater eccentric RFD ($r(s) = -0.44$, $p = 0.02$, 95% CI [$-0.71$ to $-0.07$]). The isometric RFD showed no correlation with any performance parameters.

**Conclusions:** A faster 10-yard sprint time was only correlated with a greater concentric RFD, whereas a smaller $COD_{deficit}$ was only correlated with a greater eccentric RFD. Overall, these results provide insights into the association between the acceleration and deceleration performance and RFDs according to muscle contraction type, which could help in the creation of effective training methods for improving acceleration and deceleration performance.

## INTRODUCTION

Physical movement involves a continuous cycle of acceleration and deceleration. In particular, sports activities demand rapid acceleration and deceleration, especially when changing directions. Rapid acceleration and deceleration require a considerable change in velocity, which is governed by impulse (force × time). Hence, rapid acceleration and deceleration necessitates the application of a substantial force over a brief period. Therefore, rapid force generation may play an important role in acceleration and deceleration performance.

The ability for rapid force production is measured as a slope of the force–time curve called the rate of force development (RFD) (*Maffiuletti et al., 2016*). Traditionally, the ability for rapid force production has been assessed during single-joint static tasks using a dynamometer. More recently, however, it has been evaluated during multi-joint dynamic or static movements using ground reaction force data. For instance, previous studies have measured RFDs during squat jumps (*Haff et al., 1997*; *Earp et al., 2011*; *González-Badillo, Jiménez-Reyes & Ramírez-Lechuga, 2017*; *Martinopoulou et al., 2022*), countermovement jumps (*Haff et al., 1997*; *Earp et al., 2011*; *Barker, Harry & Mercer, 2018*; *Morris, Weber & Netto, 2020*; *Krzyszkowski, Chowning & Harry, 2020*; *Ahmadi et al., 2021*), and static (isometric) squats (*Wilson et al., 1995*; *Lum, Haff & Barbosa, 2020*). However, strict control of conditions (*e.g.*, joint angles and angular velocities) is more difficult with multi-joint tasks than with single-joint static tasks, especially for dynamic movements, such as squat jumps and countermovement jumps. In contrast, single-joint tasks do not accurately represent actual multi-joint movements, which inherently involve acceleration and deceleration. Therefore, RFDs measured during multi-joint dynamic movement tasks may be more strongly associated with whole-body movement performance than those measured during single-joint tasks.

RFDs can be measured under concentric, eccentric, and isometric contractions, with muscle (*i.e.*, muscle–tendon unit) behavior being estimated based on inferences drawn from joint-level movements. During multi-joint movements tasks, RFDs measured during squat jumps, countermovement jumps, and static (isometric) squats have been referred to as concentric (*Martinopoulou et al., 2022*; *Kozinc, Smajla & Šarabon, 2024*), eccentric (*Barker, Harry & Mercer, 2018*; *Morris, Weber & Netto, 2020*; *Krzyszkowski, Chowning & Harry, 2020*), and isometric RFDs (*Wilson et al., 1995*; *Lum, Haff & Barbosa, 2020*), respectively. RFDs measured under various contraction types have generally showed no correlation, except for concentric and isometric RFDs (*Wilson et al., 1995*). This finding implies that despite common underlying factors, such as muscle size (*Maffiuletti et al., 2016*), RFDs associated with various contraction types were not inherently interdependent. Moreover, differences in muscle strength associated with contraction types (*i.e.*, concentric, eccentric, isometric) are not identical between individuals, and the correlation between maximal muscle strength and RFD has not always been strong and may not be

statistically significant (*McGuigan & Winchester, 2008*; *De Witt et al., 2018*). As such, previous studies have considered RFD and maximal strength as independent variables (*West et al., 2011*; *Dos'Santos et al., 2017*; *Brady et al., 2020*). Considering the aforementioned findings, we can surmise that the RFD ratios across different contraction types could also vary among individuals and that RFDs cannot be estimated from maximal strength. Hence, RFDs for each contraction type and each individual should be examined. Based on joint angle changes, the lower limb muscles (*i.e.*, muscle–tendon units) are believed to shorten during acceleration, whereas eccentric muscle contractions are expected to occur during deceleration. Therefore, acceleration and deceleration performance may be independently related to concentric and eccentric RFDs, respectively.

Several studies have investigated the relationship between acceleration and deceleration performance and RFDs measured during multi-joint movement tasks (*Wilson et al., 1995*; *Tillin, Pain & Folland, 2013*; *Wang et al., 2016*; *Morris, Weber & Netto, 2020*). Accordingly, evidence suggests that better acceleration performance, measured during linear sprinting, was correlated with greater concentric RFD but not eccentric or isometric RFD (*Wilson et al., 1995*). Conversely, change of direction performance was correlated with isometric RFD measured by an isometric midthigh pull (*Wang et al., 2016*). However, the total time required for the change of direction cannot be used for assessing deceleration performance given that change of direction tests include both deceleration and acceleration. Hence, change of direction performance may better serve as an indicator of acceleration performance rather than deceleration performance, given its strong association with sprint times (*Nimphius et al., 2013*). For the precise quantification of deceleration performance, *Nimphius et al. (2013)* introduced the concept of a change of direction deficit ($COD_{deficit}$), which can be defined as the required deceleration time for a deceleration calculated by subtracting the linear sprint time from the total time required for a change of direction. To the best of our knowledge, no study has been conducted on the relationship between deceleration performance (*i.e.*, $COD_{deficit}$) and RFDs. Hence, the association between deceleration performance and eccentric RFD, as well as the independent relationship between acceleration and deceleration performance are concentric and eccentric RFDs, respectively, remain obscure.

The current study therefore aimed to investigate the relationship between RFDs measured under different muscle contraction types and acceleration and deceleration performance evaluated through linear sprint running and $COD_{deficit}$, respectively. We hypothesized that a faster linear sprint time would correlate with a greater concentric RFD and that a smaller $COD_{deficit}$ would correlate with a greater eccentric RFD.

## MATERIALS AND METHODS

### Subjects

A total of 28 healthy individuals (13 males and 15 females; age: 21 ± 2 years; height: 1.66 ± 0.09 m; body mass: 65 ± 10 kg) were included as study subjects. After explaining the purpose of the research and the potential risks involved with participation, written informed consent was secured from each subject. This study was approved by the Waseda

University Ethics Committee on Research with Human Subjects (approved number: 2020-301) and was performed in accordance with the Declaration of Helsinki.

## Design

To examine the relationship between the ability for rapid force production and acceleration and deceleration performance, squat jump (SJ), countermovement jump (CMJ), and isometric squat (ISQ) tests were conducted to measure concentric, eccentric, and isometric RFDs, respectively. Acceleration and deceleration performance was evaluated using the 10-yard sprint time and $COD_{deficit}$ calculated from the pro-agility test time, respectively.

## Procedure

Before obtaining the measurements, a 10-min warm-up (aerobic exercises and dynamic stretches) was performed. The SJ, CMJ, and ISQ tests were then conducted to obtain the RFDs. To assess acceleration and deceleration performance, the subjects performed a linear sprint and a pro-agility test. The subjects were instructed on the proper performance of the movements. These tests were completed in a single day and in a randomized order, with sufficient rest periods between each test. The subjects were allowed to practice before the test to familiarize themselves with the tests.

## SJ and CMJ

The SJ is a vertical jump without countermovement. The ground reaction force measured using a force plate (9287C; Kisler, Winterthur, Switzerland) was monitored over time to visually confirm countermovements during the SJ (*Martinopoulou et al., 2022*). The subjects performed SJs from a squat position with their hands on their hips and their knee joints at a 90° angle. The CMJ was performed from a standing position, with the deepest squat position set at a knee joint angle of 90°. An angle goniometer was used to ensure that the subjects' knee joint were at an angle of 90° during all trials. A string was placed at the height of the buttocks during the deepest squat position for each trial (Fig. 1). The subjects were instructed to lower their buttocks until it touched the string. For the SJ and CMJ, the subjects practiced each at 30%, 70%, and 100% of their maximum effort before the measurements and then performed the SJ and CMJ three times as fast and high as possible. When the subjects squatted too low to touch the string during the CMJ or when countermovement were detected during the SJ, they were given sufficient rest before performing another attempt until three successful trials were achieved.

## ISQ

To assess the isometric RFD, the ISQ was performed with a bar fixed to the automatic control squat rack (LIFTER, intelligent motion Gmbh, Linz, Austria). The subjects pushed the bar set on their upper trapezius muscle upward in a squat position with the knee joint at a 90° angle (Fig. 1). The subjects were instructed to apply 40 N of force to the bar before performing the maximal effort ISQ to confirm that the bar was on their shoulders (*Tillin, Pain & Folland, 2013*). The subjects received real-time visual feedback on the ground reaction force measured through a force plate (9287C; Kisler, Winterthur, Switzerland) to
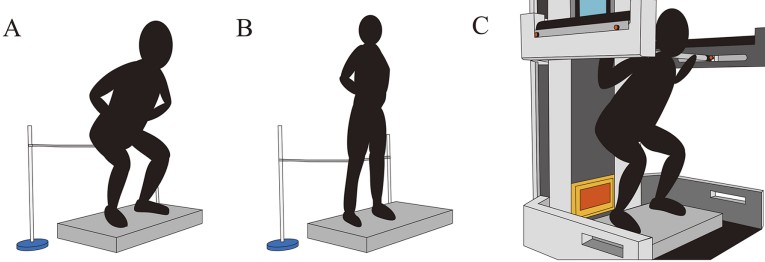

**Figure 1** **The measurement conditions.** (A) Squat jump. (B) Countermovement jump. (C) Isometric squat.

enable them to adjust their force to match a target level, which was set at the body weight of each subject +40 N. The subjects were instructed to push on the bar with maximum effort and speed, maintaining their peak effort for 3 s. After the practice session, two successful ISQ trials were conducted. If countermovement were observed, the subjects rested for approximately 2 min and repeated this until two successful trials were made. Ground reaction force data were recorded on a personal computer *via* an A/D converter (PowerLab; ADInstruments, Bella Vista, NSW, Australia) with dedicated software (LabChart version8; ADInstruments, Bella Vista, NSW, Australia) at 1,000 Hz.

## RFD calculation

The differential waveform of the time–force curve was computed as the RFD waveform. The mean concentric RFD was calculated over the time interval of 0.29 ± 0.06 s, from the onset of the SJ to the point of maximum ground reaction force (Fig. 2A). The onset of the SJ was determined as the point at which the RFD shifted from negative to over zero when tracking back from the maximum ground reaction force. The mean eccentric RFD was calculated over the time interval of 0.31 ± 0.06 s, from the point of minimum ground reaction force to the point at which the gravity center velocity reached 0 m/s (Fig. 2B) (*Harry, Barker & Paquette, 2020*). The gravity center velocity was calculated by dividing the ground reaction force by the body mass and integrating time. The mean isometric RFD was calculated over a time interval of 200 ms (*Tillin, Pain & Folland, 2013*; *Dos'Santos et al., 2017*; *Wells et al., 2018*; *Brady et al., 2020*) from the onset of the ISQ (Fig. 2C). The onset was determined as the point at which the RFD shifted from negative to over zero when tracking back from the maximum ground reaction force. The ISQ, SJ, and CMJ trials with the highest instantaneous RFD values within each time interval were used for subsequent analysis (*Mpampoulis et al., 2021*). All RFDs were normalized to body mass.

## Linear sprint test

For the linear sprint test with a standing start, the 5-yard (4.57 m) and 10-yard (9.14 m) sprint times were obtained using dual-beam phototubes (Speed Light, Swift Performance, Australia) placed at the start, 5 yards, and 10 yards. The subjects were instructed to assume a starting posture to orient their body in the advance direction and were asked to run as fast as possible to the finish line set 5-m away from the 10-yard mark (9.14 m). The trial

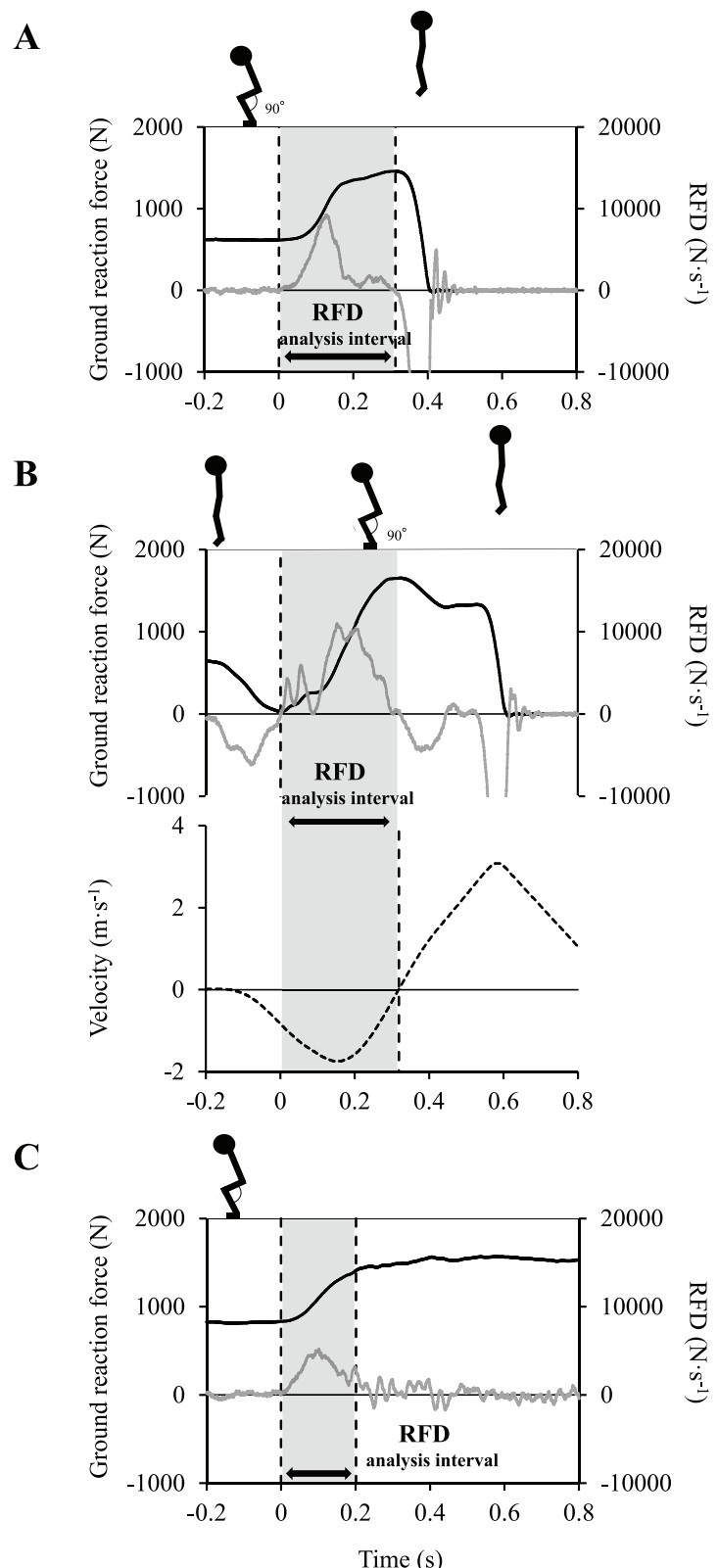

**Figure 2 Analysis time intervals of the rates of force development (RFDs).** (A) Concentric RFD. (B) Eccentric RFD. (C) Isometric RFD. Black line: ground reaction force, gray line: RFD, dashed line: velocity of center of mass.

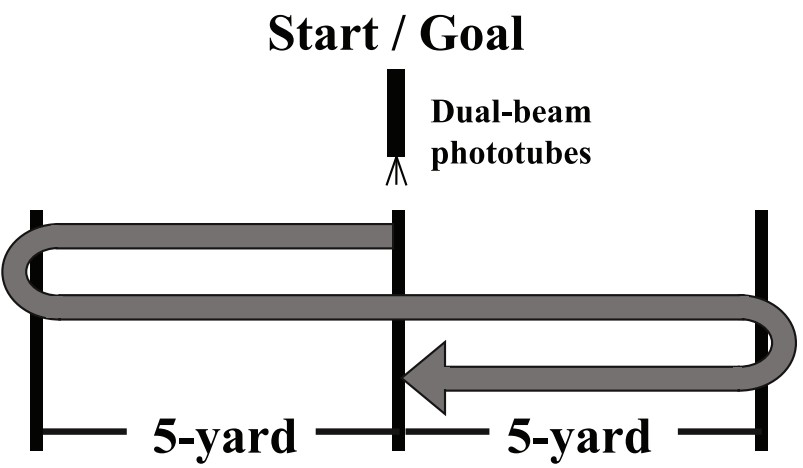

**Figure 3 Schematic presentation of pro-agility test.**

was performed twice with approximately 1 min of rest. When an obvious deceleration was observed, another trial was conducted after sufficient rest until two successful trials were achieved. The trial with the shortest 10-yard (9.14 m) sprint time was used for further analysis.

## Pro-agility test

The pro-agility test, a field assessment, involved subjects changing direction within 10 yards (9.14 m), with the central point of the 10-yard distance set as the starting and goal point (Fig. 3). The dual-beam phototubes were placed at the center point as the starting and goal points, and the change of direction point was set up 5 yards (4.57 m) in front of and behind the center point. The start posture involved the subjects facing the intended direction. We then instructed the subjects to turn 180° and run as fast as possible. The trials were performed twice and repeated if the foot failed to cross the line marking the change of direction. After sufficient rest, the trials were repeated until two successful trials were recorded. The trial with the shortest time to goal was used for analysis. The following formula was used to calculate deceleration ability: $COD_{deficit}$ = pro-agility test time − (5-yard sprint time × 2 + 10-yard sprint time).

## Statistical analysis

Data normality was confirmed *a priori* using the Shapiro–Wilk test. Correlation analyses were performed to determine the relationships between the RFDs and the 10-yard sprint time and $COD_{deficit}$. The Pearson product-moment correlation coefficient (r) was used for a combination of normally distributed dependent variables. The Spearman rank correlation coefficient (r(s)) was used for a combination of non-normally distributed variables. In all analyses, the significance level was set at 5%.

## RESULTS

Table 1 contains the statistics of the obtained data. Given that the concentric RFD and $COD_{deficit}$ were not normally distributed, relationships involving either concentric RFD or

**Table 1 Statistical variables for rates of force development and acceleration and deceleration performance.**

|  | Mean | Standard deviation | Median | Interquartile range | Maximum | Minimum | Standard error |
|---|---|---|---|---|---|---|---|
| RFD |  |  |  |  |  |  |  |
| Concentric RFD ($N \cdot s^{-1} \cdot kg^{-1}$) | 48.8 | 17.2 | 42.1 | 22.1 | 94.8 | 28.9 | 3.2 |
| Eccentric RFD ($N \cdot s^{-1} \cdot kg^{-1}$) | 71.9 | 21.9 | 74.4 | 21.3 | 116.5 | 27.9 | 4.11 |
| Isometric RFD ($N \cdot s^{-1} \cdot kg^{-1}$) | 32.6 | 14.3 | 36.2 | 22.7 | 62.2 | 8.7 | 2.7 |
| Acceleration and deceleration performance |  |  |  |  |  |  |  |
| 10-yard sprint (s) | 1.87 | 0.14 | 1.85 | 0.13 | 2.20 | 1.60 | 0.03 |
| $COD_{deficit}$ (s) | 1.00 | 0.39 | 0.98 | 0.55 | 1.77 | 0.46 | 0.75 |

**Note:**
RFD, rate of force development; 10-yard sprint, 10-yard sprint time; $COD_{deficit}$, change of direction deficit.

$COD_{deficit}$ were analyzed using Spearman's correlation coefficient, whereas other relationships (*i.e.*, between eccentric RFD and 10-yard sprint and between isometric RFD and 10-yard sprint) were analyzed using Pearson's correlation coefficient. A faster 10-yard sprint time was only correlated with a greater concentric RFD (r(s) = −0.41, *p* = 0.03, 95% CI [−0.69 to −0.03]) (Fig. 4). Meanwhile, a smaller $COD_{deficit}$ was only correlated with a greater eccentric RFD (r(s) = −0.44, *p* = 0.02, 95% CI [−0.71 to −0.07]).

## DISCUSSION

The primary findings of the present study were as follows: (1) a faster 10-yard sprint time was correlated with a greater concentric RFD but not with eccentric or isometric RFDs, (2) a smaller $COD_{deficit}$ was correlated with a greater eccentric RFD but not with concentric or isometric RFDs. Consistent with our hypotheses, these results indicate that the association between acceleration and deceleration performance with rapid force production ability is dependent on the type of muscle contraction.

Significant negative correlations were observed between 10-yard sprint time and concentric RFD (Fig. 4), corroborating the findings of a previous study (*Wilson et al., 1995*). We believe that our study has been the first to elucidate that $COD_{deficit}$ was negatively correlated with eccentric RFD but not concentric or isometric RFDs (Fig. 4). One possible explanation for the selective correlation observed between performance and RFDs could be the matching muscle contraction type between performance and RFD measurements. During short-distance acceleration sprints, such as 10-yard sprints, the lower limb joints extend during the ground contact phase of acceleration (*i.e.*, muscles contract concentrically) (*Schache et al., 2019*). During the deceleration phase of a change of direction, the primary contributing muscle groups likely undergo lengthening contractions (*i.e.*, eccentric contractions) considering joint movements (*Verheul, Harper & Robinson, 2024*).

The current study has certain limitations worth noting, with the composition of the study population being the primary concern. To recruit a sufficient number of subjects from a broad population, we invited individuals who could perform the measurement tasks without difficulty. As a result, our subjects comprised both athletes and nonathletes from various backgrounds, resulting in considerable variations in skill levels related to

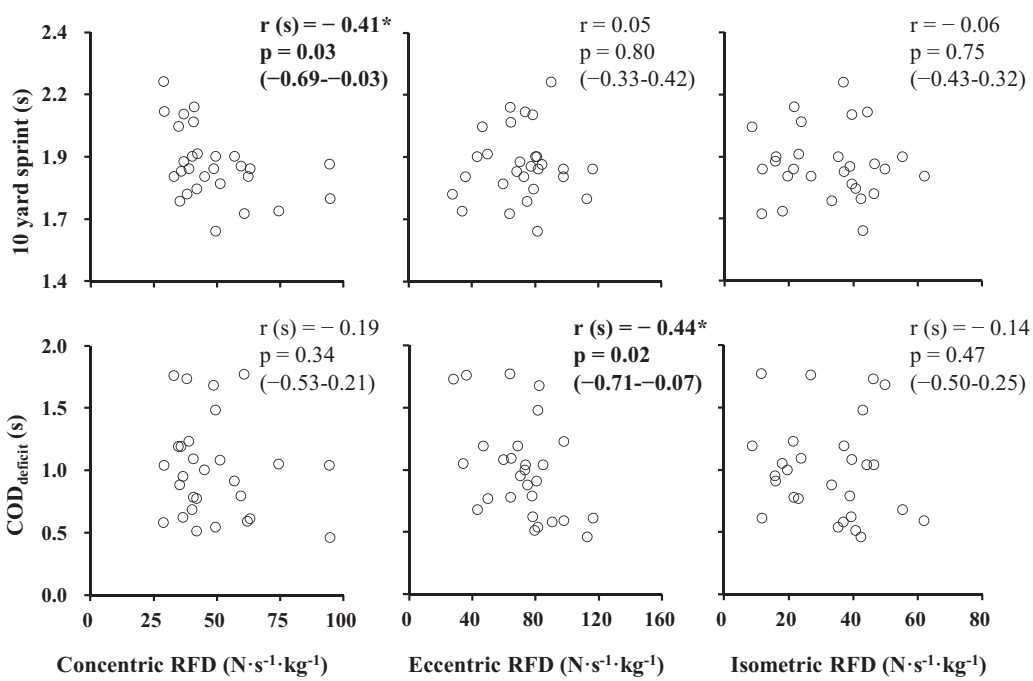

**Figure 4 Scatter plots of the rates of force development (RFDs) with 10-yard sprint time and change of direction deficit time ($COD_{deficit}$).** Pearson product-moment correlation coefficient was written as r. Spearman rank correlation coefficient was written as r(s). 95% confidence intervals are described in parenthesis. $^*p < 0.05$.

linear sprints and changes of direction. Therefore, differences in skill level may have contributed to the individual differences in performance variables (sprint time and $COD_{deficit}$). However, the supplementary data, which shows the athlete status and sex, demonstrated that the observed correlations were not simply due to differences between subgroups (males *vs.* females and athletes *vs.* nonathletes). Another limitation is the method for measuring RFDs. Precise control of the measurement conditions (*e.g.*, angular velocities) and underlying muscle–tendon behavior during multi-joint dynamic movements is quite difficult. Nevertheless, the current study confirmed significant relationships between acceleration or deceleration performance and RFDs, emphasizing the existence of a muscle contraction type-specific association between acceleration and deceleration performance and rates of force development.

The present results highlight the importance of rapid force production on acceleration or deceleration performance and its dependence on muscle contraction type, which may facilitate the development of individualized training regimens. Specifically, training aimed at enhancing concentric or eccentric RFD could be beneficial for individuals focusing on improving their acceleration or deceleration capabilities. More specifically, improvement in concentric RFD can be achieved through explosive concentric contraction exercises that demand high levels of RFD (*MacKenzie, Lavers & Wallace, 2014*), such as weightlifting exercises (*e.g.*, hang power clean starting from a midthigh position (*Comfort, Allen & Graham-Smith, 2011*)) and possibly jumping deadlifts performed with a trap bar.

Moreover, previous research has shown that fast eccentric squats (*Bogdanis et al., 2018*) and consecutive countermovement jumps may improve eccentric RFD. Although a previous study (*Andersen et al., 2010*) reported a training-induced improvement in RFD, a limited number of studies have explored the muscle contraction type-specific effects of training on RFD (*Weng et al., 2022*). Future studies are required to clarify this aspect to develop training strategies that effectively enhance acceleration and deceleration performance.

## CONCLUSIONS

The current study examined the relationship between acceleration and deceleration performance and RFDs measured under different muscle contraction types. This study has been the first to elucidate that the acceleration or deceleration performance was specifically related to concentric or eccentric RFD. This muscle contraction type-dependent association between acceleration or deceleration performance and the ability for rapid force production offers insights that could guide the creation of effective training regimens.

### Funding
This work was supported by JSPS KAKENHI Grant Number JP21K11427. The funders had no role in study design, data collection and analysis, decision to publish, or preparation of the manuscript.

### Grant Disclosures
The following grant information was disclosed by the authors:
JSPS KAKENHI: JP21K11427.

### Competing Interests
The authors declare that they have no competing interests.

### Author Contributions
- Hikaru Kurosaki conceived and designed the experiments, performed the experiments, analyzed the data, prepared figures and/or tables, authored or reviewed drafts of the article, and approved the final draft.
- Ema Tsubota performed the experiments, authored or reviewed drafts of the article, and approved the final draft.
- Motoki Katsuge performed the experiments, authored or reviewed drafts of the article, and approved the final draft.
- Kosuke Hirata conceived and designed the experiments, authored or reviewed drafts of the article, and approved the final draft.
- Kuniaki Hirayama conceived and designed the experiments, authored or reviewed drafts of the article, and approved the final draft.

## Human Ethics

The following information was supplied relating to ethical approvals (*i.e.*, approving body and any reference numbers):

This study was approved by the Waseda University Ethics Committee on Research with Human Subjects (approved number: 2020-301).

## Data Availability

Raw data is available in the Supplemental Files.

## Supplemental Information

Supplemental information for this article can be found online at http://dx.doi.org/10.7717/peerj.19862#supplemental-information.

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
