# Peer review of "Muscle contraction type-specific association of acceleration and deceleration performance with rates of force development"

_PeerJ, doi:10.7717/peerj.19862_

## Round 0.1 · original submission · Major Revisions

The reviewers were generally supportive of this submission. However, the potential problems with the study design indicate the need for a Major Revision. As an Academic Editor, I outline my general review concerns below.

1. Clarify the theoretical framework
• Although the empirical results are clearly described, the manuscript lacks an explicit theoretical context.
• Please articulate how your study fills a knowledge gap relative to existing muscle‑force theories (e.g., Hill‑type, posture‑ and velocity‑dependence of force generation) and explain how these ideas guided your hypotheses and interpretation.
2. Statistical design and control of error rates
• It is difficult to determine how many parametric vs. non‑parametric regression tests were run and how the family‑wise error rate was managed.
• Provide a concise description of: the total number of planned statistical comparisons; criteria used to choose Pearson or Spearman coefficients; any adjustments for multiple testing (e.g., Holm, Benjamini–Hochberg, or pre‑registration‑based justification).
3. Terminology: “joint‑based rate of force development”
• The Introduction currently mixes joint‑level and muscle‑level concepts (e.g., concentric and eccentric RFDs) without justification.
• Either (a) supply a rationale for using joint‑based RFD terminology or (b) rewrite the Introduction so that it consistently frames RFD at the muscle level.
4. Language and readability
• Reviewers found the content intelligible, but the manuscript would benefit from careful language editing (grammar, syntax, and word choice).
• In particular, avoid over‑using the abbreviation “rate of force development (RFD).” Spelling out the term where contextually helpful will improve flow and reduce cognitive load on readers.

·

Basic reporting

This manuscript describes a study investigating the relationship between a multi-joint measure of rate of (ground reaction) force and measures of acceleration and deceleration during sprinting and changing sprint direction, respectively. A clear rationale for the study, with sufficient overview of existing knowledge is presented. The methods are, except for a few details (see below), clearly described. Reporting of data was generally in line with the aims and hypotheses of the study. However, some additional implicit hypotheses were tested (correlation with total time of pro-agility test and correlations to peak GRF). This should be justified in the introduction or these results should be removed. This then also applies to the discussion of these results in the discussion section.

Abstract: Please include a conclusion based on the results of this study and not only an implication of the findings.

Experimental design

- The authors calculate the mean RFD for correlation with performance measures, but peak ground reaction forces. Why not using means or peaks for both parameters. This seems to be a more fair comparison.
- The selection of the subjects was not justified. Why were both athletes and non-athletes included? Is is also not clear why gender and athlete status was represented in the figures.
- It was not clear how the 90 degrees knee angles was confirmed.
- On line 155 the application of a low-pass filter is decribed. Such a filter is generally used to remove high frequency noise, but the goal with such a high cutoff frequency (500 Hz) is not clear.
- ln 165; why was 200 ms used?

Validity of the findings

As stated above the comparison between RFD and PF with regard to their correlation with performance measures requires a rationale. However, one not discussed explanation is that for RFD a mean over a certain time was used while for the GRF a peak amplitude was selected. The authors also imply that peak GRF is a measure for maximal strength (e.g. line 247), but it is not clear if that is actually measured during this task.

Additional comments

- The authors use the term RFD to indicate rate of changes in ground reaction force data. In other studies not the force, but joint moments are measured. Rate of torque development or rate of GRF development seems more appropriate.
- The authors impose different tasks to assess different muscle contraction types. However, they do not make clear if this defined at the muscle-tendon unit level or the muscle fascicle/fiber level. Several studies have shown fascicle shortening or isometric fascicle conditions during MTU lengthening. It should be stated if the authors mean MTU or fascicle when talking about muscle contraction type.

Minor comments:
ln 74; are abound
ln 85. declaration should be deceleration
Table 1; why were the results for the 5-yard time not included?

·

Basic reporting

• The English was unambiguous, and professional English was used throughout. Below are line-specific comments and recommendations:
o Line 25: “liner” should be “linear”
o Line 42: consider providing more context to why you are talking about changes in velocity as it relates to acceleration/deceleration, as it had not been previously mentioned
o Line 43: consider changing “to” to “for”
o Line 44: should it read “generation of force”
o Lines 50-54: are the citations describing multi-joint movements to measure RFD?
o Line 61: should it read “RFDs can be measured…”
o Line 73: consider rephrasing
o Line 76: should it read “measured in linear sprinting"
o Line 78: should it read “by an isometric midthigh pull…”
o Line 94: consider changing “shorter” to “faster” throughout the text when describing quicker times in linear speed; consider changing “shorter” in the CODdeficit to “smaller” or “lesser”
o Line 103: perhaps more details on the non-athletes; how were subjects recruited for the study?
o Line 117: consider changing to “warm-up”
o Line 119: were the subjects instructed on the maneuvers?
o Line 223: consider stating that there were two main findings; consider that after the “1), 2)…” the next letter should not be capitalized; a comma and not a period should be used before introducing the next number (i.e., RFDs, 2)….”
o Line 258: consider changing acceleration/deceleration, as it looks like you may be dividing the two
o Limitations section: did you consider discussing the separate analyses for the athletes versus the non-athletes from Figure 3 in the text?
o Line 267: is there a specific type of weight training exercises that would enhance capabilities?
o Line 268: should read as “deadlifts”
o Line 269: do you have a reference for the use of the accentuated squat+ for improving eccentric RFD?
• The manuscript conforms to professional standards of courtesy and expression.
• Literature references were sufficient for field background/context provided; 8 of the articles were from the last 5 years.
• The manuscript included sufficient introduction and background to demonstrate how the work fits into the broader field of knowledge. Relevant prior literature was appropriately referenced.
• The authors used professional article structure, figures, and tables. Raw data was shared.
• The structure of the article should conform to an acceptable format of ‘standard sections’ (see our Instructions for Authors for our suggested format). Significant departures in structure should be made only if they significantly improve clarity or conform to a discipline-specific custom.
• Figures were relevant to the content of the article, of sufficient resolution, and appropriately described and labeled. However, additional figures of the testing conditions would provide richer context.
• All appropriate raw data was made available in accordance with the Data Sharing policy.
• The findings were self-contained with relevant results to hypotheses.
• The submission is ‘self-contained,’ and represents an appropriate ‘unit of publication’, and includes results relevant to the hypothesis.

Experimental design

• The submission is original primary research within the Aims and Scope of the journal.
• The research question was well defined, relevant, and meaningful. It is stated how research fills an identified knowledge gap.
• The submission clearly defines the research question, which is relevant and meaningful. The knowledge gap being investigated was identified, and statements were made as to how the study contributes to filling that gap.
• The researchers performed to a high technical & ethical standard.
• Methods are described with sufficient detail and information to replicate.

Validity of the findings

• Impact and novelty were assessed and clearly stated.
• All underlying data were provided; they are robust, statistically sound, & controlled.
• The data on which the conclusions are based is provided in an acceptable discipline-specific repository. The data is statistically sound and controlled.
• Conclusions are well stated and linked to the original research question. and limited to supporting results.
• The conclusions are appropriately stated and connected to the original question investigated and are limited to those supported by the results.

Additional comments

• The authors researched a topic that is of importance to practitioners in sport. The findings of this study could be translated into practical applications in the field of strength and conditioning in sport.

---

## Round 0.2 · Minor Revisions

Thank you for the substantial effort in revising the submission. All the major concerns were adequately addressed. Please resolve the question about signal filtering to move forward.

·

Basic reporting

Comments were addressed adequately.

Experimental design

- On line 155 (first version) the application of a low-pass filter is described. Such a filter is generally used to remove high frequency noise, but the goal with such a high cutoff frequency (500 Hz) is not clear.

The authors respond to this comment with stating that this was also used by others. After reading the Balshaw et al. (2016) paper it becomes clear that the low pass filter with a high cutoff frequency is used by them to only minimally smoothen the signal. Also that this is needed for an accurate assessment of the onset time of the contraction, but not for the assessment of the RFD. To differentiate a signal, for example to calculate the rate of force change, generally, a low-pass filter with a low frequency (<10 Hz) is applied. A power spectral density analysis will probably yield very little power in the high frequency range. Apparently, the raw signal is already quite smooth. At least, the waveforms in Figure 2 look very smooth, and thus low-pass low frequency filtering may not be needed. Although, it would still be safer to apply one to make sure a high peak due to noise is found. Please make the description of the goal of the filtering more accurate.

Validity of the findings

Comments were addressed adequately.

·

Basic reporting

The authors have addressed reviewer comments and revised the section to meet professional standards. Results are self-contained and align with the stated hypotheses.

Experimental design

The authors have revised the section to align with the journal's Aims and Scope, more clearly defining a relevant and meaningful research question that addresses an identified knowledge gap, better supported by appropriate background and literature references. Methods are now described with sufficient detail for reproducibility, complemented by well-structured figures, tables, and shared raw data, all meeting professional standards of clarity and courtesy.

Validity of the findings

The authors have revised the manuscript to align with the journal's Aims and Scope, articulating a clear, significant research question that addresses a specific knowledge gap, supported by a comprehensive background and relevant literature. The technical standards are now clearer. The figures and tables have been enhanced. Conclusions are concisely stated.

Additional comments

no comment

---

## Round 0.3 · accepted · Accept

Thank you for adequately addressing all reviewers' concerns and comments. The manuscript is ready for publication.

·

Basic reporting

All comments have been addressed adequately.

Experimental design

All comments have been addressed adequately.

Validity of the findings

All comments have been addressed adequately.